# Biotransformation of Chromium (VI) via a Reductant Activity from the Fungal Strain *Purpureocillium lilacinum*

**DOI:** 10.3390/jof7121022

**Published:** 2021-11-29

**Authors:** Juan Fernando Cárdenas González, Ismael Acosta Rodríguez, Yolanda Terán Figueroa, Patricia Lappe Oliveras, Rebeca Martínez Flores, Adriana Sarai Rodríguez Pérez

**Affiliations:** 1Unidad Académica Multidisciplinaria Zona Media, Universidad Autónoma de San Luis Potosí, Carretera San Ciro de Acosta Km. 4.0, Ejido Puente del Carmen, Ríoverde C.P. 79617, Mexico; juan.cardenas@uaslp.mx; 2Laboratorio de Micología Experimental, Centro de Investigación y de Estudios de Posgrado, Facultad de Ciencias Químicas, Universidad Autónoma de San Luis Potosí, Av. Dr. Manuel Nava No. 6, Zona Universitaria, San Luis Potosí C.P. 78320, Mexico; iacosta@uaslp.mx; 3Laboratorio de Microbiología, Parasitología y Toxicología de Alimentos, Facultad de Enfermería, Universidad Autónoma de San Luis Potosí, Av. Niño Artillero No. 130, Zona Universitaria, San Luis Potosí C.P. 78320, Mexico; yolandat@uaslp.mx; 4Departamento de Botánica, Instituto de Biología, Universidad Nacional Autónoma de México, Ciudad Universitaria, Delegación Coyoacán, Ciudad de México C.P. 04510, Mexico; lappe@ib.unam.mx (P.L.O.); rebecam@ib.unam.mx (R.M.F.)

**Keywords:** *Purpureocillium lilacinum*, bioremediation, heavy metal, biotransformation, Cr (VI) reduction, enzymes

## Abstract

Industrial effluents from chromium-based products lead to chromium pollution in the environment. Several technologies have been employed for the removal of chromium (Cr) from the environment, including adsorption, ion-exchange, bioremediation, etc. In this study, we isolated a Cr (VI)-resistant fungus, *Purpureocillium lilacinum,* from contaminated soil, which could reduce chromium. We also characterized a reductant activity of dichromate found in the cellular fraction of the fungus: optimal pH and temperature, effect of enzymatic inhibitors and enhancers, metal ions, use of electron donors, and initial Cr (VI) and protein concentration. This study also shows possible mechanisms that could be involved in the elimination of this metal. We observed an increase in the reduction of Cr (VI) activity in the presence of NADH followed by that of formate and acetate, as electron donor. This reduction was highly inhibited by EDTA followed by NaN_3_ and KCN, and this activity showed the highest activity at an optimal pH of 7.0 at 37 °C with a protein concentration of 3.62 µg/mL.

## 1. Introduction

Chromium is an important metal, which is required in several industrial processes, such as textile manufacturing, leather tanning, chrome plating, wood processing, and ceramics, dyes, paints, and pigment manufacturing [1]. The waste from these processes is dumped in rivers and aquifers, leading to environmental pollution and the biomagnification of chromium in the environment. Exposure to chromium causes several toxic effects, namely dermatitis, allergies, cancer, mutations, and teratogenic effects, which have been attributed to hexavalent chromium ions [Cr (VI)] [2]. Chromates are highly toxic because they are very strong oxidizing agents and water-soluble in a wide pH range; and the structure of chromate oxyanions is like that of sulfate and phosphate ions, which increases their possibility of crossing biological membranes and generating a reducing intracellular environment. The reduction of Cr (VI) inside the cell could generate Cr (V), Cr (III), and reactive oxygen species (ROS). The molecular mechanisms that are responsible for DNA damage involve interactions between intracellular Cr (III) and DNA, proteins, and ROS production [3]. On the other hand, trivalent chromium [Cr (III)] is an essential micronutrient, which is indispensable in glucose metabolism and to the proper functioning of insulin, lipids, and proteins in mammals [4]. This is reflected by a decrease in triglyceride and cholesterol levels due to the activation of enzyme reactions and increase in muscle mass [2]. 

For the reduction and removal of Cr (VI) from wastewater and soil, many physical/chemical methods, such as adsorption, ion-exchange, precipitation, electrodialysis, reverse osmosis, and biological reduction methods (bioremediation) have been developed [5]. Many of these technologies are costly and pose other disadvantages like incomplete removal/reduction, tons of sludge production, high reagent consumption, and the requirement for a large amount of energy [6]. However, microorganism-based reduction or removal of heavy metals is an efficient and more effective alternative, including dilute solutions [7]. Most microorganisms that have been studied for chromate reductase activity include bacteria, like *Pseudomonas*, *Bacillus*, *Arthrobacter*, and *Escherichia coli* [8,9,10], yeast, like *Candida maltose* [11] and *Candida utilis* [12], and a few fungi, such as *Hypocrea tawa* [13], *Aspergillus* sp. [14], and *Penicillium* sp. [15]. Chromate reductase has been characterized (physicochemical and biochemical properties) from crude extracts of microorganisms for the better elucidation of their role and mechanism in the reduction of Cr (VI) in the membrane fraction, or intra and extracellularly [16]. In *Stenotrophomonas maltophilia*, a protein of 25 kDa was isolated from the cell free extract, a reductase responsible for Cr (VI) reduction [17]. In another study, a chromate reductase from *Thermus scotoductus* SA-01 was purified to homogeneity, which contained a flavin mononucleotide as cofactor [18]. A chromate reductase isolated reported in *Escherichia coli* had four subunits with flavin mononucleotide as cofactor [19]. *Thermoanaerobacter thermohydrosulfuricus* BSB-33 was also reported to possess a chromate reductase and its genome sequence was studied [20].

In 2013, it was reported that chromate reduction is mediated by a four-electron transfer chromate reductase [21] in *Pseudomonas stutzeri* L1 and *Acinetobacter baumannii* L2; Cr (VI) reduction activity was identified in the soluble fractions for higher efficiency of Cr (VI) bioreduction [22]. In *Bacillus subtilis*, a membrane bound enzyme in association with phosphate groups is involved in chromate reduction [23]. In *Aspergillus niger*, chromate reductase was isolated from the soluble fraction and was involved in Cr (VI) reduction and removal, and intracellular accumulation of Cr (III), and different groups of the cell wall of this fungus participate in binding of the metal [24]. Activity of chromate reductases is not specific to chromates; they have many other activities involving degradation of organic or inorganic substrates, and thus, they are capable of bioremediation of contaminated effluents [16]. Recently, it has been reported that oxidation processes with ROS like sulfate radicals, hydroxyl ions, superoxides, and singlet oxygen, represent a good strategy for the degradation of organic compounds in water, like the removal of bisphenol A by superoxide radical and singlet oxygen generated from peroxymonosulfate activated with Fe0-montmorillonite [25], the removal of tetrabromobisphenol A by α-Fe_2_O_3_ [26], the removal of dissolved organic matter by peroxymonosulfate activation of Fe-Al [27], the degradation of pollutants by peroxydisulfate non-radical/radical activation over layered CuFe oxide [28], and the degradation of phenol by immobilization of *Sphingomonas* sp. GY2B in polyvinyl alcohol-alginate-kaolin beads [29].

Bioremediation of heavy metals like Cr (VI) by microorganisms is a viable and environmental-friendly approach to reduce environmental pollution. Though there are several techniques for removal of hexavalent chromium, the application of chromate reductases for the reduction of Cr (VI) to Cr (III) is a more efficient and effective strategy, which has not been explored for the elimination of this metal from wastewater in aerobic and anaerobic conditions (cytosolic and bound membrane forms) [16]. 

Therefore, in this study, we analyzed whether the fungus *Purpureocillium lilacinum* isolated from a Cr (VI)-contaminated environment, which is Cr (VI)-resistant, shows chromate reductase activity. We also partially characterized this activity and located the cellular fraction.

## 2. Materials and Methods

### 2.1. Fungus Isolation and Culture Conditions

The Cr (VI)-resistant microorganism was isolated through open and prolonged exposure of petri dishes with solid culture medium from a place near the Faculty of Chemical Science, San Luis Potosi, México, which was contaminated with industrial vapors. Were placed on petri dishes containing modified Lee’s Minimal Medium [30], prepared in triple ionized water with 0.20% MgSO_4_, 0.50% (NH_4_)_2_SO_4_, 0.25% KH_2_PO_4_, 0.50% NaCl, 0.25% glucose, which contained 500 mg/L K_2_Cr_2_O_7_, and the pH of the medium was adjusted to 7.0 with 100 mMol/L of sodium-phosphate buffer. The cultures were incubated for 7 days at 28 °C. All strains found by prolonged and open exposure were identified by its macro and micro-morphological characteristics, such as the color of the strain, the shape, and diameter of the spores, choosing only one strain of interest. Fungal culture grown in thioglycolate broth was used as the primary inoculum to keep the strain growing and available. All experiments were performed in triplicate along with their respective controls. The purified fungal strain was obtained after serial transfer on PDA (Potato Dextrose Agar) and was stored in the Fungal Collection of the National Herbarium of Mexico (MEXU) under the accession number MEXU 27801. The fungal strain was preserved in WA (Water Agar) (0.2%) at 4 °C, and in 30% glycerol-potato dextrose broth (PDB) at 30 °C in the Laboratory of Mycology C006, Institute of Biology, National Autonomous University of Mexico.

### 2.2. Morphological and Phylogenetic Identification of the Fungal Strain 

Morphological characteristics (macro and micromorphological) and sequence analysis of the internal transcribed spacer (ITS) 1-5.8S-ITS2 region were used for the taxonomic identification of the fungal strain. The fungal strain was grown on different culture media (MEA malt: extract-peptone-glucose-agar, CYA: Czapek-yeast extract agar, G25N: glycerol-nitrate-agar) and incubated at different temperatures (5, 25, and 37 °C) for one week [31,32,33]. The cultures were observed daily, growth rate was measured, and colony description and morphological structures were examined by brightfield microscopy. For sequence analysis, total genomic DNA was extracted after 5 days of incubation from a PDA culture, using the FT71415 Rapid Fungal Genomic DNA isolation kit (Bio Basic Inc., Markham, Canada) according to the manufacturer’s protocol. The ITS-5.8S region was amplified with the universal primers ITS5 (5′-GGAAGTAAAAGTCGTAACAAGG-3) and ITS4 (5′-TCCTCCGCTTATTGATATGC-3′) [34]. A PCR reaction (25 µL containing 12.5 µL GoTaq^®^ Master Mix (Promega, Madison, Wisconsin, USA) using 1.25 µL of each primer (10 pm/µL) and 5 ng of genomic DNA was performed in a Thermal Cycler Gernandt 2720 (Applied Biosystems, Singapore). The PCR program consisted of an initial denaturing step at 94 °C for 1 min, followed by 30 cycles of 1 min at 94 °C, 2 min at 58 °C, 1 min at 72 °C, and a final extension step for 5 min at 72 °C [35]. Sequencing was done at the Laboratory of Molecular Biology of Biodiversity and Health, Institute of Biology, UNAM, México, and then edited using the BioEdit Program v 7.0.5. The edited sequence was aligned against GenBank sequences using BLASTN program [36].

### 2.3. Preparation of Cell Free Extracts (CFE) 

The CFE of the fungus was prepared using a previous method [15]. Briefly, culture suspensions of the fungus *P. lilacinum* were grown for 5 days in 200 mL thioglycolate broth (pH 7.0), harvested by centrifugation for 10 min at 3000× *g* at 4 °C, and the cell pellet (10 mL) was washed twice with 100 mM potassium phosphate buffer (pH 7.0, at 4 °C) and resuspended in the same buffer. The fungal cells were incubated at 4 °C and disrupted with an Ultrasonic Mini Bead Beater Probe (Densply) with 15 cycles of 1 min each, and the obtained cell suspension was centrifuged at 3500× *g* for 20 min at 4 °C, for the removal of cell wall and unbroken cells. The supernatant was centrifuged at 14,000× *g* for 60 min at 4 °C to obtain the working fraction, which was maintained in the same buffer and used as crude reductant activity.

### 2.4. Partial Characterization of Reductant Activity 

Reductant activity reduction was estimated as described previously using a standard curve of Cr (VI) 0–30 mM [37]. The concentration of the metal fraction obtained was assayed at 37 °C in a final 1000 µL reaction volume, which contained 50 µL NADH (10 mM), 200 µL of sodium-phosphate buffer (pH 7.0, 100 mM), 500 µL of Cr (VI) (0–30 mM), and 250 µL of the CFE. This volume (1.0 mL) was used for all experiments. Assay conditions were kept constant with a reaction time of 6 h and 37 °C unless stated otherwise. The reductant activity was measured at 37 °C and different pH values using several buffers (100 mM phosphate citrate, pH 5.0; 50 mM phosphate, pH 6.0–8.0, and 50 mM TRIS-HCl, pH 8–9) and at different temperatures (20–60 °C) at the optimum pH. The effects of different metal cations on the reductant activity were also analyzed using solutions of different salts (10 mM), electron donors (1 mM), and different inhibitors (1 mM).

### 2.5. Reductant Activity Assay and Protein Determination 

Cr (VI) concentration was determined by a colorimetric method, wherein a pink-violet colored complex is formed with the diphenylcarbazide (DPC) in acid solution [38], and the total protein by Lowry´s method [39]. Unit enzyme activity for dichromate reductase was defined as the amount of enzyme that reduces 1 µg of Cr (III)/min at 37 °C. Specific activity was defined as the unit of dichromate reductase activity (U) per mg of protein in the CFE. 

### 2.6. Characterization of Cell Fractions by Fourier-Transform Infrared (FTIR) Spectroscopy

The cell fraction of interest, mixed membrane fraction in this case, was placed in the presence and absence of Cr (VI) for 7 days. The cells were dried in an oven at 80 °C until dry, and then the dry biomass was pulverized in a mortar. For FTIR analysis, 10 mg of the mixed membrane fraction was mixed with 10 mg of KBr and the FTIR spectrum was determined before and after contact with Cr (VI) in a Perkin Elmer infrared (500–4000 cm^−1^) spectrophotometer using KBr to make films, which serve for functional groups analysis of 20 mg/L and control without Cr (VI).

## 3. Results and Discussion

### 3.1. Molecular Characterization of the Fungal Strain

The isolated fungal strain was identified by its macromorphological and micromorphological characteristics as *Paecilomyces*. The molecular identification of the fungal strain was carried out by nucleotide analysis of the ITS-5.8S region using the BLAST program, which verified the strain as *Paecilomyces lilacinum*, currently known as *Purpureocillium lilacinum*, based on 100% sequence homology of the isolated strain with the sequence of similar fungal strains (Table 1). The nucleotide sequence has been submitted in GenBank with accession number KR025539.

### 3.2. Localization of Reductant Activity

The highest reductant activity, considered as a likely chromate reductase activity, was observed in the mixed membrane fraction (Figure 1a,b), the results showed that with the addition of NADH^+^ as electron donor, reductant activity increased approximately six times at 28 °C, pH 7.0, and stirring at 100 rpm. These findings are in accordance with the results obtained for *B. subtilis* [23], *Enterobacter cloacae* [40], *Pseudomonas fluorescens* [41], *Staphylococcus aureus*, and *Pediococcus pentosaceus* [42], where a higher reductant activity related to chromium reduction was also associated with the membrane fraction. However, these results differ from those obtained for fungi *Penicillium* sp. [15] and *A. niger* [24], bacteria *S. maltophilia* (17), and yeast *Trichosporon asahii* and *Rhodotorula mucilaginosa* [43], wherein the reductant activity was reported in the CFE (soluble fraction).

### 3.3. Effect of pH on Reductant Activity

To determine the optimum pH for reductant activity, this was measured at different pH. At low pH, the changes could be due to the interaction between proton (H^+^) and chromium in the cell wall ligands, such as carboxyl, phosphate, and amino groups, because chromium uptake is protein and enzyme mediated, and pH could affect the degree of ionization of proteins affecting chromium uptake. The crude dichromate reductase of *P. lilacinum* exhibited higher activity in sodium-phosphate buffer pH 7.0 (Figure 2), and these results are like those reported for the fungi *Penicillium* sp. [15], *A. niger*, and *A. parasiticus* [44] and the yeasts *R. mucilaginosa* [43] and *Pichia jadini* M9 [45]. Other studies have reported stability of dichromate reductase activity between pH 6.5 and 7.5 for *E. coli* CFE [46], between pH 5.0 and 8.0 for *Bacillus* sp. [47], and at pH 6.0 for yeast *Trichosporon asahii* [43].

### 3.4. Effect of Temperature on the Reductant Activity 

The effect of temperature (28–60 °C) on the reductant activity was also evaluated, and the maximum activity was observed at 37 °C (Figure 3). This result is comparable to that reported in other studies. However, the reductase activity was not significantly altered between 28 and 60 °C. This was unlike what was expected due to denaturation of proteins in heat (5 min at 100 °C, data not shown). This result is comparable with that reported in other studies: Cr (VI) reductase activity of *Penicillium* sp. (15) with an optimal temperature of 37 °C, while *Pseudomonas* sp. G1DM21 [8], *R. mucilaginosa* [43], *A. niger*, *A. parasiticus* [44], *E. coli* [46], and *Bacillus* sp. CFEs [47] showed higher activity at 30 °C. On the contrary, yeast *T. asahii* [43] and bacteria *Pseudomonas putida* CFE [48] were more resistant to high temperature, and the enzyme activity was stable up to 50 °C.

### 3.5. Effect of Initial Chromium Concentration on Crude Reductant Activity 

The enzyme kinetics was analyzed between 2–10 mg/100 mL chromium concentration, the activity decreased with increasing initial chromium concentration (Figure 4). The specific activity decreased from 2800 with 2 mg/100 mL of Cr (VI) to 75 with 10 mg/100 mL of Cr (VI), and from the lowest concentration an increase in enzyme activity is observed, reaching 2 mg/100 mL, apparently the maximum reaction rate. Above this concentration, there is a significant decrease in the activity. This result is like that reported for *P. jadinii* M9 where removal was observed after 2 h for initial concentrations 0.4 and 0.7 mM, and after 5 h for 1 mM [45]. For *Candida tropicalis* [49], when the initial Cr (VI) ion concentration was increased from 25 to 100 mg/L, and for *Bacillus methylotrophicus* when the initial concentration was increased from 0.25 to 0.50 mM [50], the percentage removal of metal ions decreased. However, for *Ochrobactrum* sp. Cr-B4, the specific activity of the enzyme increased with an increase in initial concentration of Cr (VI), up to 350 mM [51].

### 3.6. Effect of Different Metal Cations on the Crude Reductant Activity

The reductant activity of *P. lilacinum* showed a significant decrease in the presence of all cations (Figure 5), and the least inhibitory effect was observed in the presence of Hg^2+^, Ca^2+^, and Na^+^, whereas the highest inhibitory effect was observed in the presence of Mg^2+^ and Cu^2+^. However, the effect of Hg^2+^ observed in this study was different from that observed in previous studies wherein Hg^+2^ strongly inhibited the hexavalent chromate reductase activity: in *Pseudomonas* sp. G1DM21 [8], *Penicillium* sp. [15], and *Bacillus firmus* KUCr 1 [52]. The effect of Cu^2+^, which had the most inhibitory effect on reductant activity, also differed from that reported for *B. firmus* KUCr 1, as an activating effect on chromate reductase activity was observed [52]. All other ions that were tested, like Na^+^, Mg^2+^, and Fe^3+^, showed an inhibitory effect on the Cr (VI) reductase activity at different levels, and this result agrees with the effects reported for *Penicillium* sp. [15], *Bacillus* sp. [47], *Ochrobactrum* sp. Cr-B4 [51], and *Arthrobacter crystallopoietes* [53]. Some metal cations serve as cofactors for reductant reactions, but no increase in the reductant activity was observed in this study.

### 3.7. Effect of Different Inhibitors on the Crude Reductant Activity 

The dichromate reductase activity of the mixed membrane fraction of *P. lilacinum* was also evaluated and significantly inhibited by all inhibitors (Figure 6). Respiratory chain inhibitors azide and cyanide caused a 50% and 40% decrease in dichromate reductase activity, respectively. Azide and cyanide inhibit aerobic chromate reduction by *Penicillium* sp. [15] and *B. subtilis* [40] and create an approximately 50% reduction in the membrane associated chromate reductase activity of *Shewanella putrefaciens* MR-1 [54]. These inhibitors affect de novo protein synthesis and the respiratory chain intermediates, where Cr (VI) serves as a terminal electron acceptor [8]. Other inhibitors like 2-mercaptoethanol affect disulfide links, which are essential in maintaining the protein structure, and therefore, cause denaturation of the reductase protein. Hence, we observed a 30% inhibition in the enzyme activity like that reported for *Ochrobactrum* sp. Cr-B4 [51]. However, 2-mercaptoethanol did not affect the reductant activity in case of *Penicillium* sp. [15].

Protease inhibitors such as EDTA that chelate metal ions, like zinc, lead, and calcium, strongly inhibited the reductant activity in our study, similar to that reported for the chromate reductase activity of *Penicillium* sp. [15] and *Ochrobactrum* sp. Cr-B4 [51]. However, no loss of reductant activity was reported in the case of *Bacillus methylotrophicus*, which suggests that chromate reductase of this microorganism does not require metal ions as a cofactor for its inherent catalytic activity [50].

### 3.8. Effect of Different Carbon Sources as Electron Donors on the Crude Reductant Activity 

The reductant activity did not increase significantly with different carbon sources as electron donors (Figure 7), which was different from the results observed in a previous study on *Pseudomonas* sp. G1DM21 [8], wherein an increase of 25%, 21%, and 14% was observed in dichromate reductase activity in the presence of acetate, citrate, and succinate, respectively. Glucose acts as an electron donor and increases Cr (VI) reduction in the case of *Bacillus* sp. isolated from chromium landfill [55]. Our results are in accordance with the studies that have reported NADH-dependent reductant reduction of Cr (VI) under aerobic conditions, e.g., in the case of *Pseudomonas* sp. G1DM21 [8], *Penicillum* sp. [15], *E. coli* ATCC 33456 [46], *Bacillus* sp. [47], and *P. putida* [48]. In the case of *A. rhombi-RE*, sodium pyruvate was the most effective electron donor [9]. According to a study on alkaliphilic *B. subtilis*, the oxidation of NADH donates an electron to the chromate reductase enzyme, and this electron is transferred to Cr (VI), forming Cr (V) as the intermediate, which finally accepts two electrons from other organic substances to produce Cr (III) [23].

### 3.9. Effect of Different Concentration of Protein on the Reductant Activity

We also analyzed the effect of different protein concentrations, between 3.62 and 7.25 µg/mL (250–500 µL of the cellular fractions), on the crude reductant activity (Figure 8) and observed increased protein concentration and decreased reductant activity, but more data are required for correct interpretation of these results. However, this result is different from that reported for *Exiguobacterium* sp. KCH5, which shows a high K_M_ value of crude chromate reductase (200 µM) and a relatively low affinity of this enzyme for Cr (VI) [43].

### 3.10. Effect of Cr (VI) on IR Region of the Subcellular Fraction

Table 2 shows that the values for the peaks corresponding to the hydroxyl groups, amides I and amides II, have not undergone significant changes or modifications. This may be because in both cases these groups are involved in the reduction of Cr (VI) and later convert to acid in the case of hydroxyl groups, in the case of amides I and II, and it is suggested that the reduction could be due to the presence of proteins [23,56,57,58,59].

### 3.11. Biotransformation of Cr (VI) to Cr (III)

Of all cell fractions analyzed, it was observed that the fraction with the greatest capacity to biotransform toxic chromium [Cr (VI)] to a non-toxic form [Cr (III)] was the mixed membrane fractions. During experimentation, the samples were left for 15 days at 37 °C and a green precipitate was observed (Figure 9). The yellow color of the samples was transformed to green color with time in comparison to the control due to the reduction and/or transformation of Cr (VI) in solution. After 15 days of work with the mixed membrane fraction, these were placed in refrigeration (4 °C). This color change may be due to the mixed membrane fractions that synthesize essential enzymes required for the accumulation and reduction of Cr (VI) to Cr (III), like hexavalent chromium reduction by *R. mucilaginosa* and *T. asahii* cells [43]. When we analyzed electrochemical speciation, Cr(OH)_3_ was found, which corresponds to the Cr (III) ions. This observation confirms the biotransformation of Cr (VI) to Cr (III) by a new reductant activity.

## 4. Conclusions

The current study elucidated the localization and characterization of a very efficient reductant activity of a dichromate reductase from *P. lilacinum*. Chromium reduction by the fungus was evaluated at a laboratory scale at an optimum temperature of 37 °C, pH 7.0, and 100 rpm with an initial concentration of 2 mg/100 mL Cr (VI) as potassium dichromate. The optimum activity was enhanced in the presence of 0.1 mM NADH, but other carbon sources as electron donors showed no substantial effects. With 0.1 mM of different cation metals, like Na^+^, Mg^2+^, Fe^3+^, Ca^2+^, Cd^2+^, Cu^2+^, and Hg^2+^, the latter showed the least decrease in activity. The results of this study suggest the potential of the enzyme in the quick reduction of Cr (VI) under laboratory conditions with no stress, which is important in the field of bioremediation, since most of the reports related to the study of the chromate reductases of microorganisms are of bacteria, and very concern few fungi. In addition, *P. lilacinum* is commonly isolated from soil, decaying vegetation, insects, nematodes, and as a laboratory contaminant and can be cultured easily in vitro in large quantities. The lack of adverse effects for the environment and/or other living beings from this fungus is also an advantage of using *P. lilacinum* for Cr (VI) mitigation from the environment.

## Figures and Tables

**Figure 1 jof-07-01022-f001:**
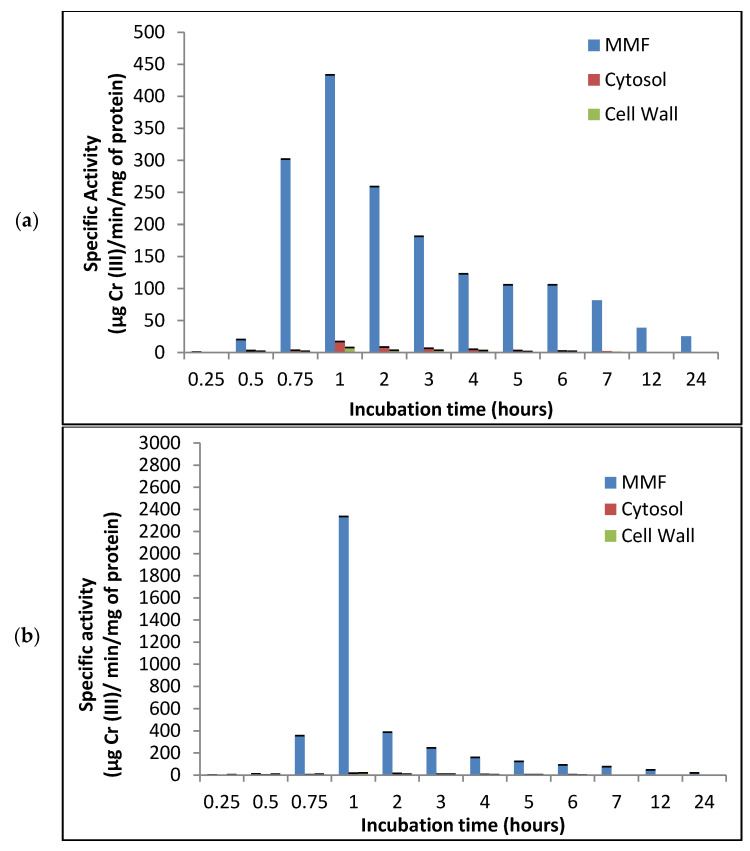
Reductant activity in different cell fractions of the fungus *P. lilacinum* at different incubation times. 28 °C, pH 7.0, 100 rpm. (**a**) Without NADH+ as electron donor, (**b**) With NADH+ 10 mM.

**Figure 2 jof-07-01022-f002:**
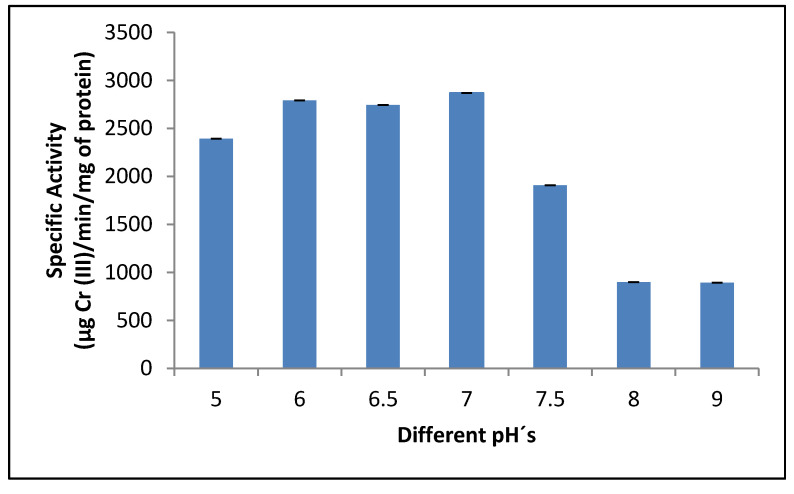
Effect of pH on crude reductant activity in the mixed membrane fraction of *Purpureocillium lilacinum* determined in different buffers (pH 5.0–9.0) at 37 °C, 1 h of incubation, and 100 rpm.

**Figure 3 jof-07-01022-f003:**
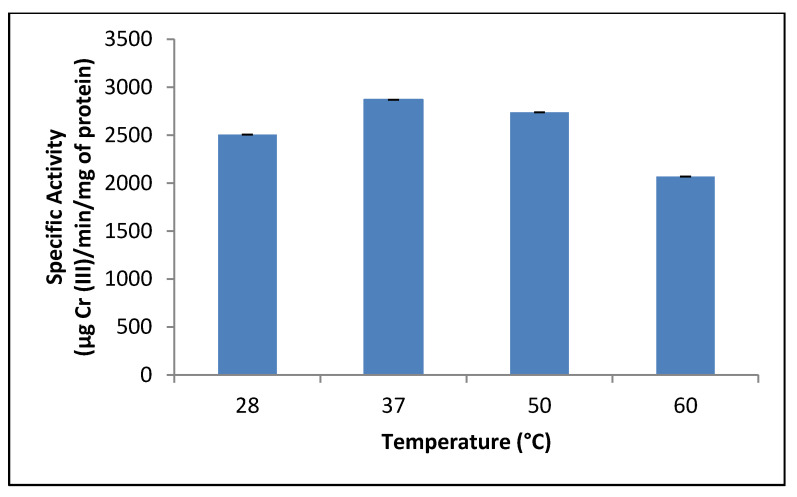
Effect of temperature on crude reductant activity in mixed membrane fraction of *Purpureocillium lilacinum*. pH 7.0, 1 h of incubation, and 100 rpm.

**Figure 4 jof-07-01022-f004:**
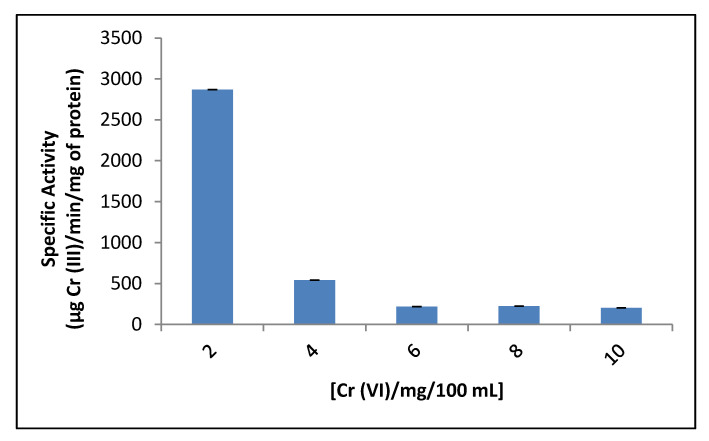
Effect of different initial concentrations of Cr (VI) on crude reductant activity in the mixed membrane fraction of *Purpureocillium lilacinum*. 1 h of incubation, pH 7.0, 100 rpm, and 37 °C.

**Figure 5 jof-07-01022-f005:**
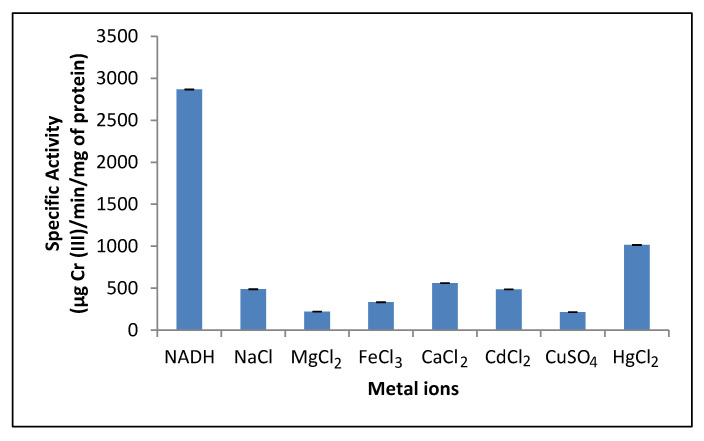
Effect of metal ions on crude reductant activity in the mixed membrane fraction of *Purpureocillium lilacinum*. 1 h of incubation, pH 7.0, 100 rpm, and 37 °C.

**Figure 6 jof-07-01022-f006:**
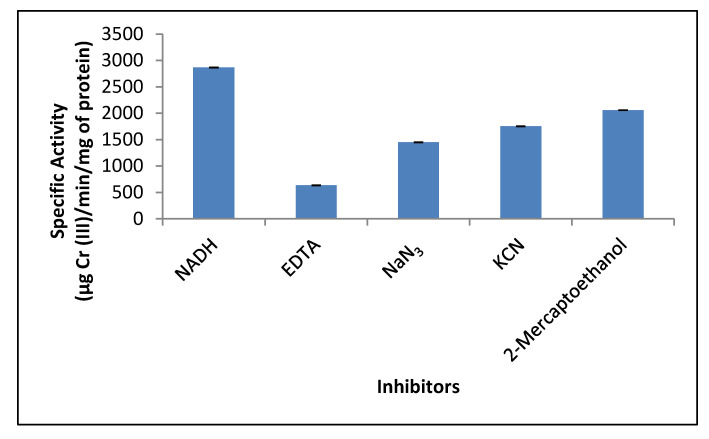
Effect of inhibitors on crude reductant activity in mixed membrane fraction of *Purpureocillium lilacinum*. 1 h of incubation, pH 7.0, 100 rpm, and 37 °C.

**Figure 7 jof-07-01022-f007:**
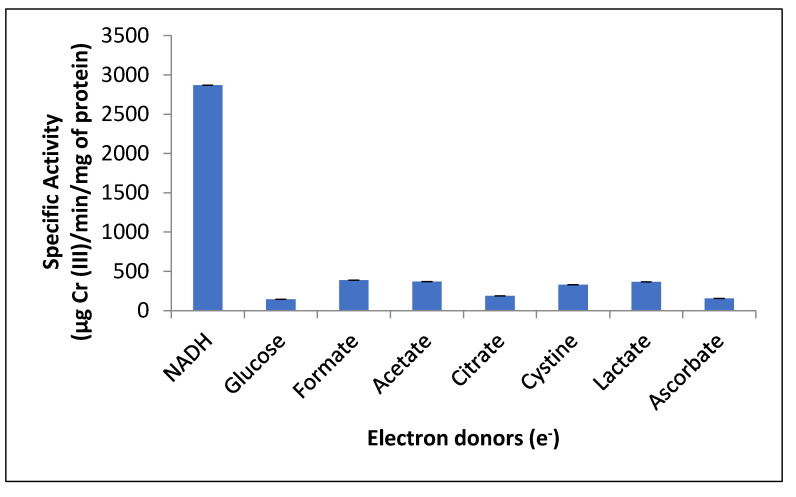
Effect of electron donors on crude reductant activity in the mixed membrane fraction of *Purpureocillium lilacinum*. 1 h of incubation, pH 7.0, 100 rpm, and 37 °C.

**Figure 8 jof-07-01022-f008:**
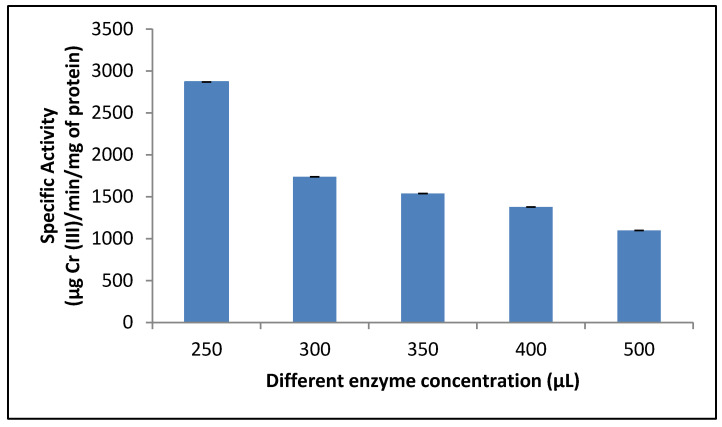
Effect of different enzyme concentrations on crude reductant activity in the mixed membrane fraction of *Purpureocillium lilacinum*. 1 h of incubation, pH 7.0, 100 rpm, and 37 °C.

**Figure 9 jof-07-01022-f009:**
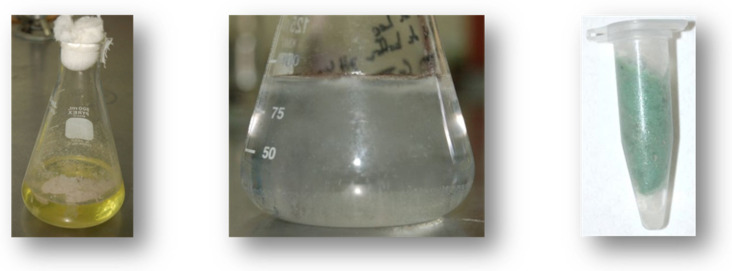
Gradual conversion from Cr (VI) to Cr (III) with green precipitate observed in the mixed membrane fraction of *Purpureocillium lilacinum* after 15 days at 37 °C with 20 mg/L Cr (VI).

**Table 1 jof-07-01022-t001:** BLASTN analysis of the ITS-5.8S sequence of the strain 001JFC with that of other similar fungi (NCBI database).

Microorganisms	Accession Number	Identity (%)
*Purpureocillium lilacinum* 001JFC	KR025539	100
*Paecilomyces lilacinus* LTBF 007-1	GQ229080	100
*Purpureocillium lilacinum* M1447	KC157713	100
*Paecilomyces lilacinus* SY45B-a	HM242264	100
*Purpureocillium lilacinum* M3905	KC157751	99
*Purpureocillium lilacinum* M3748	KC157748	99
*Purpureocillium lilacinum* M3516	KC157741	99

**Table 2 jof-07-01022-t002:** FTIR spectrum analysis of the subcellular fraction before and after treatment with 20 mg/L of Cr (VI).

	Wave Numbers (cm^−1^)	
Functional Groups before and after Treatment	Cell Fraction Native	Cell Fraction Treated with 20 mg/L
O-H stretching vibration from polysaccharides and proteins	3405.7	3410.6
CH_3_ asymmetric stretching from proteins	2910.3	2922.5
Amide (I) group from protein (C=O)	1652.1	1650.4
Amide (II) group (N-H) + (C-N) from protein	1540.4	1540.4
Amide (III) group (C-N)	1402.5	1360.7
C-O, SO and/or PO stretching vibrations	1072.7	1064.6
Groups CrO_4_^−2^	-------	500–900

## Data Availability

The sequence of the fungal isolate *Purpureocillium lilacinum* 001JFC is available in the Genbank database under the accession number KR025539.

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
