# Peer review of "Biotransformation of Chromium (VI) via a Reductant Activity from the Fungal Strain Purpureocillium lilacinum"

_jof, 2021, doi:10.3390/jof7121022_

Round 1

Reviewer 1 Report

Dear Authors

The manuscript is really very interesting, that is the biotransformation of Chromium (VI) to Chromium (III) through Chromate Reductase from the fungal strain Purpureocillium lillacinum, shows a real possibility of reducing the dangerous hexavalent Cr. Other attempts often move the problem eg. accumulating the chromium that passes from the soil to the plants, etc. but they don't solve the problem. However, the authors must clarify the following points:
1- Did they carry out real applications or in the field? 2- The incubation reported in Fig. 9 (line 412) shows that the incubation parameters are 15 days at 4 ° C with 20 mg/L Cr (VI), however the activity was tested respect to various temperatures did not include + 4 ° C therefore the activity at + 4 ° C is not included if it is lower or higher than the other temperatures. Chromate Reductase tests at lower temperatures (+ 4 ° C, + 10 ° C, etc.) should be carried out because are closer to the realities in the field; 3- The authors should by means of SEM and/or atomic absorption (AA) where the Cr III is stored in the fungus Purpureocillium lillacinum (eg vacuoles, wall, etc.).

Best Regards

Author Response

Reply to reviewer 1

Corrections were made and marked in green

The manuscript is really very interesting, that is the biotransformation of Chromium (VI) to Chromium (III) through Chromate Reductase from the fungal strain Purpureocillium lillacinum, shows a real possibility of reducing the dangerous hexavalent Cr. Other attempts often move the problem eg. accumulating the chromium that passes from the soil to the plants, etc. but they don't solve the problem.

R= You are absolutely correct, many research groups carry out oxidation-reduction mechanisms through physical-chemical methodology and with this there is the formation of large amounts of compounds that are still pollutants, which continue to have valence states that continue to leave them as toxic.

Our investigations look for microorganisms that change those toxic compounds to non-toxic or, where appropriate, less toxic.

1.-Did they carry out real applications or in the field?

Yes, we carried out this type of study previously, experiments were made with soil contaminated with a certain concentration of chromium (VI), this bioremediation study had real characteristics.

The information comes in the following reference:

Cárdenas-González, J. F. and Acosta-Rodríguez, I. 2010. Hexavalent chromium removal by a Paecilomyces sp. fungal strain isolated from environment. Bioinorganic Chemistry and Applications 2010: 1–6.

2- The incubation reported in Fig. 9 (line 412) shows that the incubation parameters are 15 days at 4 ° C with 20 mg/L Cr (VI), however the activity was tested respect to various temperatures did not include + 4 ° C therefore the activity at + 4 ° C is not included if it is lower or higher than the other temperatures. Chromate Reductase tests at lower temperatures (+ 4 ° C, + 10 ° C, etc.) should be carried out because are closer to the realities in the field

R= We apologize for the inconvenience, it is an error in writing, the working temperature was 37 ° C for 15 days, and after these tests the samples were kept at a refrigeration temperature (4 ° C).

A paragraph has already been added in the work on this topic

3- The authors should by means of SEM and/or atomic absorption (AA) where the Cr III is stored in the fungus Purpureocillium lillacinum (eg vacuoles, wall, etc.).

At the moment we do not have these methodologies available since the one that was available suffered breakdown and failures, but the idea is to identify this part through SEM, as soon as the pandemic allows it and the equipment works, we will try in the future. Excellent observation, thank you very much.

*the modifications were made in the uploaded document

Reviewer 2 Report

This manuscript describes the Biotransformation of Chromium (VI) by a via the fungal strain, Purpureocillium lilacinum. Although this biotransformation has been previously described in other bacterial and fungal strains, the process is interesting due to the toxic environmental effects of Cr(VI) salts, that are highly oxidant species.

However, the main concern about accepting the manuscript is the description of a new enzyme with “ Chromate Reductase” activity. Authors could re-consider the biotransformation of Cr(VI) without proposing this transformation to a new  enzyme, but this choice should avoid so many parts of the manuscript that assume the existence of an enzyme. The enzyme is never characterized, and the pattern of the system hardly reconciles with an enzyme.

The characterization of an enzyme should include some essential properties, such as the characterization of the  substrates and products, kinetics properties, rate-dependence vs substrate concentration versus enzyme amount and so on. Unfortunately, the information described in the manuscript is very confusing and somehow suggests that there is no enzyme, but a sort of chemical redox reactions between dichromate and the organic biomass contained at the fungal cells extracts. The manuscript would be rejected at the current state, and it could be re-considered after a complete re-consideration of the experimental data. The following points would be addressed, considering that there are the main points, as the manuscript is full of minor points that should be corrected.

Material and Methods: 2.3 Preparation of cell free extracts (CFE).

Is that preparation used throughout the paper? The section establishes at the end sentence that this preparation is used as crude chromate reductase (line 154). However, most of the sections are obtained using MMF (Mixed Membrane Fraction), as enzymatic extract. This preparation is not described, and it should do. The “chromate reductase” assay should be also described at Material and Methods, and a blank control using (di)chromate) plus NADH in the absence of fungal extract should be included.

Is this enzyme catalysing the reaction of dichromate (oxidant) with NADH (reductant) as substrates? The characterization of the reaction and its proposal is essential. The point about the oxidant species is very important. At pH<7, the Cr(VI) main species is dichromate, Cr2O7-2. However, most of the manuscript is written assuming chromate. The term dichromate is only mentioned at the conclusion. This is unacceptable, as this point affect the number of electrons transferred by the “chromate reductase” activity (6 electrons, not 3). All paragraphs concerning electrons exchange should consider that point. In addition, the oxidation of NADH + H+ to NAD+ needs 2 electrons, not just one (see for instance line 354 of the manuscript)..

The last sentence at the abstract: is not acceptable. It has no sense to establish that the highest activity of the enzyme is obtained using a protein concentration of 3.62 μg/mL of crude extract. It is not possible to inhibit any enzymatic reaction by increasing the amount of the catalyzer. This affect section 3.9 and figure 8.  It is unacceptable the statement that the  data indicate that the enzyme has high affinity for the substrate and a low KM (data not shown). If the authors have a Km value, they should report it, as well as the Lineweaver-Burke graphs, dependence with substrate concentration and so on. This is an essential data to report a “chromate reductase” enzyme. Moreover, related to Figure 4, that pattern cannot be described as enzyme saturation. The activity is not saturated, it is inhibited. If so, it is an inhibition by excess of substrate. Authors should consult any textbook about enzymology. Usually, this pattern indicate that the system is not dependent of one enzyme. Suggestion, NADH oxidation could be easily monitoring at 340 nm.

Expression of substrate concentrations: The concentration of the substrate, Cr(VI) is a mess, difficult for comparison. See for instance 3.5: At the beginning, the concentration is given as mg/100 mL, but also mM (0.4 to 0.7 mM),  also as 25-100 mg/L and finally as 0.25-0.5 mM in the same paragraph. Note that the conversion of mg/L to mM depends of considering chromate or dichromate. Authors should try to be ordered, as the manuscript is long and it should facilitate comparison among different paragraphs.

Figures 5, 6 and 7: Is NADH added in all samples, or are the supposed cations, electron donors and inhibitors added instead of NADH? Description is unclear. Details such as concentration of the agents should be added. Some of the electron donors at Figure 7 can  hardly act as electron donors, and inhibitors  at Figure 6 are inhibitors of cytochromes and metal-enzymes, but they are not of a number of NADH dehydrogenases. They are not just protease inhibitors (see line 334), as this study is not studying a protease activity.

Section 3.10 should be omitted. The IR data are not really related to the manuscript, It is detachable material. In addition, the assignation of the bands showed at Table 2 are interpretable and opinionable, as well as the shifts in the band peaks. In turn, the IR spectra are not shown. Data only demonstrate that there is protein and (di)chromate salt, but this is obvious.

Section 3.11 is baffling. The whole paper is written considering 1 h as the standard reaction time for Cr(VI) reduction, but this section reports some data after 15 days. This observation strongly suggest that Cr(VI) reduction is a spontaneous process. It does not confirm that the biotransformation of Cr (VI) to Cr (III) is due to a new chromate reductase enzyme (as authors state, see line 408).The enzyme cannot be synthesised in this acellular system (mixed membrane fraction).

Minor points

Avoid the expression NADH+, references should be ordered by order of appearance, use always salt notation (i.e formic acid and acetate at the abstract instead of formate and acetate), use always italics for latin names (line 66, Stenotrophomonas maltophilia)

Author Response

Reply to reviewer 2

The corrections were made and marked in yellow and were also answered as follows

However, the main concern about accepting the manuscript is the description of a new enzyme with “Chromate Reductase” activity. Authors could re-consider the biotransformation of Cr (VI) without proposing this transformation to a new enzyme, but this choice should avoid so many parts of the manuscript that assume the existence of an enzyme. The enzyme is never characterized, and the pattern of the system hardly reconciles with an enzyme.

R.- This was corrected in the text. We report a activity of chromate reductase.

2.- The characterization of an enzyme should include some essential properties, such as the characterization of the substrates and products, kinetics properties, rate-dependence vs substrate concentration versus enzyme amount and so on. Unfortunately, the information described in the manuscript is very confusing and somehow suggests that there is no enzyme, but a sort of chemical redox reactions between dichromate and the organic biomass contained at the fungal cells extracts. The manuscript would be rejected at the current state, and it could be re-considered after a complete re-consideration of the experimental data. The following points would be addressed, considering that there are the main points, as the manuscript is full of minor points that should be corrected.

R.- This was corrected in the text

3.- Material and Methods: 2.3 Preparation of cell free extracts (CFE).

Is that preparation used throughout the paper? The section establishes at the end sentence that this preparation is used as crude chromate reductase (line 154). However, most of the sections are obtained using MMF (Mixed Membrane Fraction), as enzymatic extract. This preparation is not described, and it should do. The “chromate reductase” assay should be also described at Material and Methods, and a blank control using (di)chromate) plus NADH in the absence of fungal extract should be included.

R.- This was corrected in the text:

The initial experiments were carried out on the crude extract, in order to determine if there was enzymatic activity, because if there was no activity, the project was not followed, and as the result was positive, the following experiments were carried out, including the determination of the activity in the different fractions analyzed and / or isolated, obtaining a higher enzymatic activity in MMF

The highest enzymatic activity, considered as a likely chromate reductase activity, was observed in the mixed membrane fraction (Figure 1a and 1b), the results showed that with the addition of NADH+ as electron donor, chromate reductase activity increased approximately 6 times at 28°C, pH 7.0, and stirring at 100 rpm

4.- Is this enzyme catalysing the reaction of dichromate (oxidant) with NADH (reductant) as substrates? The characterization of the reaction and its proposal is essential. The point about the oxidant species is very important. At pH<7, the Cr(VI) main species is dichromate, Cr2O7-2. However, most of the manuscript is written assuming chromate. The term dichromate is only mentioned at the conclusion. This is unacceptable, as this point affect the number of electrons transferred by the “chromate reductase” activity (6 electrons, not 3). All paragraphs concerning electrons exchange should consider that point. In addition, the oxidation of NADH + H+ to NAD+ needs 2 electrons, not just one (see for instance line 354 of the manuscript)..

R.- This was corrected in the text:

All paragraphs concerning electrons exchange should consider that point. In addition, the oxidation of NADH + H+ to NAD+ needs 2 electrons, not just one. This is correct, but in the literature, there are reports in which the enzymatic activity, use NADH y not NADH + H, how electron donors, with good results, and below we add some examples:

  1. Desai, C., Jain, K., Madamwar, D. Hexavalent Chromate Reductase activity in cytosolic fraction of Pseudomonas sp G1DM21 isolated from Cr (VI) contaminated from landfill. Process Biochem. 2008, 43, 713-721.

 The cell-free extracts (CFE) reduced 90% of 100 μM Cr(VI) in 120 min. The Km and Vmax determined for chromate reductase activity in the CFE were 175 μM Cr(VI) and 1.6 μmoles/min/mg of protein, respectively, the Km and Vmax determined in the presence of 0.5 mM NADH were 150 μM Cr(VI) and 2.0 μmoles/min/mg of protein, respectively

2.- Studies on biological reduction of chromate by Streptomyces griseus.

Poopal AC, Laxman RS.J Hazard Mater. 2009 Sep 30;169(1-3):539-45. doi: 10.1016/j.jhazmat.2009.03.126. Epub 2009 Apr 5.PMID: 19410364

The enzyme was constitutive and did not require presence of chromate during growth for expression of activity. Chromate reduction with cell free extract (CFE) was observed without added NADH. However, addition of NAD(P)H resulted in 2-3-fold increase in activity.

3.- Rahman Z, Thomas L. Chemical-Assisted Microbially Mediated Chromium (Cr) (VI) Reduction Under the Influence of Various Electron Donors, Redox Mediators, and Other Additives: An Outlook on Enhanced Cr(VI) Removal. Front Microbiol. 2021 Jan 28;11:619766. doi: 10.3389/fmicb.2020.619766. PMID: 33584585; PMCID: PMC7875889.

In an investigation, addition of Cr(VI) shifted the amounts of fermentation products toward more oxidative form, i.e., acetate (∼2.5 times) (Sharma, 2002). The plausible reason of this shift toward acetate formation after Cr(VI) exposure is likely to secure more NADH molecule that can be channeled toward bioreduction process. Mass balance reaction also supported the reason, which derived more loss of NADH molecules by butyrate and lactate formation than acetate synthesis (Sharma, 2002).

4.- Opperman DJ, Piater LA, van Heerden E. A novel chromate reductase from Thermus scotoductus SA-01 related to old yellow enzyme. J Bacteriol. 2008 Apr;190(8):3076-82. doi: 10.1128/JB.01766-07. Epub 2008 Feb 8. PMID: 18263719; PMCID: PMC2293266.

Enzyme activity was also dependent on NADH or NADPH, with a preference for NADPH, coupling the oxidation of approximately 2 and 1.5 mol NAD(P)H to the reduction of 1 mol Cr(VI) under aerobic and anaerobic conditions, respectively.

5.- Elangovan R, Philip L, Chandraraj K. Hexavalent chromium reduction by free and immobilized cell-free extract of Arthrobacter rhombi-RE. Appl Biochem Biotechnol. 2010 Jan;160(1):81-97. doi: 10.1007/s12010-008-8515-6. Epub 2009 Jan 23. PMID: 19165627.

The enzyme activity was optimal at a pH of 5.5 and a temperature of 30 degrees C. Among the ten electron donors screened, sodium pyruvate was the most effective one followed by NADH and propionic acid. Michaelis-Menten constant, K(m), and maximum reaction rate, V(max), obtained from the Lineweaver-Burk plot were 48 microM and 4.09 nM/mg protein/min, respectively, in presence of NADH as electron donor

6.- Desai C, Jain K, Madamwar D. Evaluation of in vitro Cr(VI) reduction potential in cytosolic extracts of three indigenous Bacillus sp. isolated from Cr(VI) polluted industrial landfill. Bioresour Technol. 2008 Sep;99(14):6059-69. doi: 10.1016/j.biortech.2007.12.046. Epub 2008 Feb 5. PMID: 18255287.

The Cr(VI) reduction by the cell-free extracts of Bacillus sp. G1DM20 and G1DM22 was maximum at 30 degrees C and pH 7 whereas, Bacillus sp. G1DM64 exhibited maximum Cr(VI) reduction at pH 6. Addition of 1mM NADH enhanced the Cr(VI) reductase activity in the cell-free extracts of all three isolates.

5.- The last sentence at the abstract: is not acceptable. It has no sense to establish that the highest activity of the enzyme is obtained using a protein concentration of 3.62 μg/mL of crude extract. It is not possible to inhibit any enzymatic reaction by increasing the amount of the catalyzer. This affect section 3.9 and figure 8.  It is unacceptable the statement that the  data indicate that the enzyme has high affinity for the substrate and a low KM (data not shown). If the authors have a Km value, they should report it, as well as the Lineweaver-Burke graphs, dependence with substrate concentration and so on. This is an essential data to report a “chromate reductase” enzyme. Moreover, related to Figure 4, that pattern cannot be described as enzyme saturation. The activity is not saturated, it is inhibited. If so, it is an inhibition by excess of substrate. Authors should consult any textbook about enzymology. Usually, this pattern indicate that the system is not dependent of one enzyme. Suggestion, NADH oxidation could be easily monitoring at 340 nm.

R.- This was corrected in the text:

--- The last sentence at the abstract: is not acceptable. It has no sense to establish that the highest activity of the enzyme is obtained using a protein concentration of 3.62 μg/mL of crude extract. It is correct, we did not mention that there is an inhibitory effect in the abstract

R.- This was corrected in the text:

--- It is unacceptable the statement that the data indicate that the enzyme has high affinity for the substrate and a low KM (data not shown). This was corrected in text

Moreover, related to Figure 4, that pattern cannot be described as enzyme saturation. The activity is not saturated, it is inhibited. R.- This was corrected in the text:

6.- Expression of substrate concentrations: The concentration of the substrate, Cr(VI) is a mess, difficult for comparison. See for instance 3.5: At the beginning, the concentration is given as mg/100 mL, but also mM (0.4 to 0.7 mM), also as 25-100 mg/L and finally as 0.25-0.5 mM in the same paragraph. Note that the conversion of mg/L to mM depends on considering chromate or dichromate. Authors should try to be ordered, as the manuscript is long, and it should facilitate comparison among different paragraphs.

In effect, different expressions of the substrate concentration are handled, our work reports it in mg / 100 mL, but some reports in the literature report it as mM, so we cannot change those, but if required we can change our values concentrations to mM

7.- Figures 5, 6 and 7: Is NADH added in all samples, or are the supposed cations, electron donors and inhibitors added instead of NADH? Description is unclear. Details such as concentration of the agents should be added. Some of the electron donors at Figure 7 can hardly act as electron donors, and inhibitors at Figure 6 are inhibitors of cytochromes and metal-enzymes, but they are not of several NADH dehydrogenases. They are not just protease inhibitors (see line 334), as this study is not studying a protease activity.

In fact, it is not a study of proteases, but in the literature these parameters that can affect the enzymatic activity are described, that is why we describe it in this way, as in section No. 4, in which some examples from the literature are cited. which can also be applied to this section.

8.- Section 3.10 should be omitted. The IR data are not really related to the manuscript, It is detachable material. In addition, the assignation of the bands showed at Table 2 are interpretable and opinionable, as well as the shifts in the band peaks. In turn, the IR spectra are not shown. Data only demonstrate that there is protein and (di)chromate salt, but this is obvious.

This was corrected in text.

9.- Section 3.11 is baffling. The whole paper is written considering 1 h as the standard reaction time for Cr(VI) reduction, but this section reports some data after 15 days. This observation strongly suggest that Cr(VI) reduction is a spontaneous process. It does not confirm that the biotransformation of Cr (VI) to Cr (III) is due to a new chromate reductase enzyme (as authors state, see line 408). The enzyme cannot be synthesised in this acellular system (mixed membrane fraction).

In effect, this observation strongly suggest that Cr(VI) reduction is a spontaneous process, but, as we are working with MMF, in which according to our data the highest enzymatic activity is found, we suggest that it may be due to the activity present in these cell fractions, which catalyzes the reaction from hexavalent chromium to trivalent chromium, although we do not rule out that There are other factors involved, because if it were a spontaneous process, there would not be so much contamination by this metal in the different places, because being spontaneous, the concentration of the metal with valence VI would frequently be reduced.

10.- Avoid the expression NADH+, references should be ordered by order of appearance, use always salt notation (i.e formic acid and acetate at the abstract instead of formate and acetate), always use italics for latin names (line 66, Stenotrophomonas maltophilia)

This was corrected in text.

*the modifications were made in the uploaded document

Reviewer 3 Report

In the manuscript entitled “Biotransformation of Chromium (VI) via a Chromate Reductase from the fungal strain Purpureocillium lilacinum” the authors identify and localize the presence of the chromate reductase enzyme in the fungus Purpureocillium lilacinum, isolated from an environment contaminated by Cr (VI), which is resistant to Cr (VI). In this context, although the introduction is clear and the results obtained are interesting, considering the possible use of this fungus in the removal of chromium from contaminated environments, the present version of the manuscript is acceptable for publication after major revisions

In paragraph 2.5 the authors defined the ‘Unit enzyme activity for chromate reductase’ as ‘the amount of enzyme that reduces 1 mMol of Cr (VI) per min at 37°C’. However, in Results section they report the values as ‘μg Cr (III)/min’. Please explain and report all results as ‘mMol of of Cr (VI) per min’.

Add SDS-PAGE analysis of protein extract studied to have an idea of proteins profile.

Paragraph 3.9: authors decided to evaluated the Km value without using a pure enzyme. How the authors calculated the total enzyme concentration? In my opinion, the Km of a protein extract can not be measured. In the extract could be present also an inhibitor responsible for decreasing of enzymatic activity when you use higher extract volume (see Figure 8). However, if the authors were able to calculate the Km, please show your data (page 11, line 368). In the same figure (8) on x axis ‘µL’ not mean concentration! I suggest to use the amount in ‘µg’ or ‘mg’;

-page 2, line 66: change ‘Stenotrophomonas maltophilia’ by ‘Stenotrophomonas maltophilia’;

-page 4, line 154: change ‘3,500 × g/20 min/4°C’ by ‘3,500 × g for 20 min at 4°C’;

-page 4, line 175: change ‘Specific activity was defined as unit chromate reductase activity per minute per mg protein concentration in the CFE’ by ‘Specific activity was defined as unit chromate reductase activity (U) per mg protein in the CFE’;

-page 5, figure 1: change ‘(μg Cr (III)/min/mg of protein)’ by ‘(U/ mg of protein). Check this in overall manuscript.

Author Response

Reply to reviewer 3

The corrections were made and marked in blue and were also answered as follows

  • In paragraph 2.5 the authors defined the ‘Unit enzyme activity for chromate reductase’ as ‘the amount of enzyme that reduces 1 mMol of Cr (VI) per min at 37°C’. However, in Results section they report the values as ‘μg Cr (III)/min’. Please explain and report all results as ‘mMol of of Cr (VI) per min’. 

It was corrected in the text,

  • Add SDS-PAGE analysis of protein extract studied to have an idea of proteins profile. 

We did not have the opportunity to carry out this experiment, as we do not have the equipment to perform the electrophoresis, it would be very important to know the protein profile of the sample, but if requested we will try to do this experiment

  • Paragraph 3.9: authors decided to evaluated the Km value without using a pure enzyme. How the authors calculated the total enzyme concentration? In my opinion, the Km of a protein extract can not be measured. In the extract could be present also an inhibitor responsible for decreasing of enzymatic activity when you use higher extract volume (see Figure 8). However, if the authors were able to calculate the Km, please show your data (page 11, line 368). In the same figure (8) on x axis ‘µL’ not mean concentration! I suggest to use the amount in ‘µg’ or ‘mg’;

  It was corrected in the text; we did not carry out the determination of the Km, because the enzyme is not pure

  • -page 2, line 66: change ‘Stenotrophomonas maltophilia’ by ‘Stenotrophomonas maltophilia’; 

It was corrected in the text

  • -page 4, line 154: change ‘3,500 × g/20 min/4°C’ by ‘3,500 × g for 20 min at 4°C’; 

It was corrected in the text

  • -page 4, line 175: change ‘Specific activity was defined as unit chromate reductase activity per minute per mg protein concentration in the CFE’ by ‘Specific activity was defined as unit chromate reductase activity (U) per mg protein in the CFE’; 

  • It was corrected in the text

-page 5, figure 1: change ‘(μg Cr (III)/min/mg of protein)’ by ‘(U/ mg of protein). Check this in overall manuscript

It was corrected in the text

*the modifications were made in the uploaded document

Round 2

Reviewer 2 Report

I have re-reviewed this version of the manuscript taking into account the reports of my two colleagues (Reviewers 1 and 3) and the reply letter from the authors. It is obvious that my opinion about this work was different from others, but I still think that this work contains important flaws from the enzymological point of view. I am convinced that fungal extracts are able to reduce Cr(VI) oxyanions to Cr(III) salts, but I had serious doubts that this reduction would be catalyzed by a specific enzyme. I have this feeling the first time that I reviewed the manuscript, and I maintained serious doubts yet. This point is essential, and it is not solved at all. However, 

The biochemical characterization of any enzyme needs to characterize without ambiguities the nature of the substrates,  the basic kinetic properties of the catalysis (Km, Vm, michaelian or non michaelian pattern, linear dependence between the reaction rate and the amount of enzyme, protein characterization by SDS-PAGE and so on). By the way, Km could be determined for impure enzymes indeed. In the first version of this manuscript, authors wrote that Km was determined (although the value was never given), and in the current second version (or in the answer to one of other reviewers) they write that Km cannot be determined in impure enzymes. The essential point for Km determination is knowing the nature of the substrates. I am sorry, but I do not know the nature of the substrates yet, especially the reductant substrate. I am grateful to authors to provide several references (until 6) concerning “chromate reductase” activity of several microorganisms. I have not read all of them carefully, but I suppose that some of them are just unprecise, similar to this one. The existence of those papers cannot be considered a justification to publish new unprecise papers. Anyway, I see that some of those papers (see no. 1, Desai et al. reported values for Km and some papers contain data concerning the supposedly reductant substrate (NADH or NAPDH) that are closer to the proposal of an enzymatic activity. Unfortunately, this is not the case.

In summary, I have no doubts that this paper describes a new microorganism able to accelerate Cr(VI) reduction under some particular conditions. I have doubts about the existence of an specific enzyme with specificity for Cr(VI) oxyanions (chromate or dichromate). The manuscript contains interesting contributions, as the phylogenetic identification of the fungal strain Purpureocillium lilacinum and undoubtedly, some of the corrections introduced in this version have improved the manuscript.

Bearing in mind these thoughts , I would suggest for the consideration of the authors (and the editor) the following changes in the current form before acceptance.

Title: Exchange the term “Enzymatic” by “Reductant” or at least “Reductase”

Abstract:

  • Exchange: “an activity of dichromate” by “a reductant activity of dichromate””
  • “formic acid and acetate” by “formic and acetic acids) or better “formate and acetate”. The anions are the predominant species at pH 7

Then, at Line 165: delete: enzymatic. From this line on, concerning the sections 2.4, 2.5 3.2, 3.3, 3.5, 3.6, 3.7, 3.8 and 3.9. The words enzyme and enzymatic should be avoided, but unfortunately the manuscript is full of these two terms. I must insist that the manuscript does not characterize any enzyme, so that it is senseless to talk about enzymatic activity throughout the paper. In turn, the pattern of the catalysis is very abnormal, suggesting that the existence of an enzyme is unlikely (i.e. the showed the highest activity is obtained with a protein concentration of 3.62 μg/mL (see abstract), assuming that concentration is referred to total protein in the fungal extract). It is hard to explain why greater amount of a supposed enzyme does not increase the activity. Catalytic system or reductant activity are the recommended terms for substituting the terms “enzyme” and “enzymatic activity”.

Author Response

Reply to reviewer 2-round 2

The authors made the changes that were suggested by reviewer 2, but, we justify some of the proposals made by the reviewer, we will work to purify and obtain data for enzyme analysis as suggested by reviewer 2, we appreciate the criticism for that in the future the work be enriched with important scientific information.

I have re-reviewed this version of the manuscript taking into account the reports of my two colleagues (Reviewers 1 and 3) and the reply letter from the authors. It is obvious that my opinion about this work was different from others, but I still think that this work contains important flaws from the enzymological point of view. I am convinced that fungal extracts are able to reduce Cr(VI) oxyanions to Cr(III) salts, but I had serious doubts that this reduction would be catalyzed by a specific enzyme. I have this feeling the first time that I reviewed the manuscript, and I maintained serious doubts yet. This point is essential, and it is not solved at all.

R.- The reviewer suggests that he doubts that the activity reported in this work is not enzymatic, but, although it was not purified, we think that with the data obtained, there are bases to suggest an enzymatic activity, in addition, in the absence of the metal there is practically no enzymatic activity, which suggests that the activity is not constitutive, and that enzymatic activity is only observed in the presence of the substrate. In addition, it has been suggested that there may be a reducing activity of Chromium (VI) in the presence of ascorbic acid, cystine, and glucose, but with this at very long times, we observed that in the presence of these compounds, after some time, the yellow solution changes to a green color and a white precipitate, indicating the reduction of chromium (VI) to chromium (III)

However, the biochemical characterization of an enzyme needs to characterize without ambiguities the nature of the substrates, the basic kinetic properties of the catalysis (Km, Vm, michaelian or non michaelian pattern, linear dependence between the reaction rate and the amount of enzyme, protein characterization by SDS-PAGE and so on). By the way, Km could be determined for impure enzymes indeed.

R.- Indeed, to obtain a biochemical characterization of the enzyme, it is required that it be pure, but it was not the primary objective of the work, since what interests us most is having efficient techniques and methodologies for the removal of metal from contaminated sites, in addition, we do not have the infrastructure necessary for its purification, especially a cold room so that in the different stages of purification, the activity is not lost and/or inactivated. We also know that it is very important to carry out electrophoresis, but we reiterate, we do not have the implements to carry it out, but if it is required we will try to re-enact it, so the enzymatic activity was only partially characterized.

In the first version of this manuscript, the authors wrote that Km was determined (although the value was never given), and in the current second version (or in the answer to one of other reviewers) they write that Km cannot be determined in impure enzymes. The essential point for Km determination is knowing the nature of the substrates. I am sorry, but I do not know the nature of the substrates yet, especially the reductant substrate. I am grateful to authors to provide several references (until 6) concerning “chromate reductase” activity of several microorganisms. I have not read all of them carefully, but I suppose that some of them are just unprecise, similar to this one. The existence of those papers cannot be considered a justification to publish new unprecise papers.

R.- We did not determine the Km, we only analyzed the effect of the amount of substrate, and we compared it with other data in the literature, in which this parameter was determined, and some reports support it, although the reviewer also suggests that they are also imprecise. , although in some of them they purify the activity, and we are not justifying our data, we simply carry out a discussion of what was obtained, to try to determine if the reported activity is equal to or better than in other Works Anyway,

I see that some of those papers (see no. 1, Desai et al. reported values for Km and some papers to contain data concerning the supposed reductant substrate (NADH or NADPH) that are closer to the proposal of enzymatic activity. Unfortunately, this is not the case.

R.- this is not the case, we only compare the data reported with our work, to strengthen and compare the data obtained by Desay, and other reports.

In summary, I have no doubts that this paper describes a new microorganism able to accelerate Cr(VI) reduction under some particular conditions. I have doubts about the existence of a specific enzyme with specificity for Cr(VI) oxyanions (chromate or dichromate).

R.- The reviewer suggests that he doubts that the activity reported in this work is not enzymatic, but, although it was not purified, we think that with the data obtained, there are bases to suggest an enzymatic activity, in addition, in the absence of the metal there is practically no enzymatic activity

The manuscript contains interesting contributions, as the phylogenetic identification of the fungal strain Purpureocillium lilacinum and undoubtedly, some of the corrections introduced in this version have improved the manuscript.

Bearing in mind these thoughts, I would suggest for the consideration of the authors (and the editor) the following changes in the current form before acceptance.

Title: Exchange the term “Enzymatic” by “Reductant” or at least “Reductase”

CORRECTIONS MADE IN THE TEXT

Abstract:

  • Exchange: “an activity of dichromate” by “a reductant activity of dichromate”

CORRECTIONS MADE IN THE TEXT

  • “formic acid and acetate” by “formic and acetic acids) or better “formate and acetate”. The anions are the predominant species at pH 7

CORRECTIONS MADE IN THE TEXT

Then, at Line 165: delete: enzymatic. From this line on, concerning the sections 2.4, 2.5 3.2, 3.3, 3.5, 3.6, 3.7, 3.8 and 3.9. The words enzyme and enzymatic should be avoided, but unfortunately the manuscript is full of these two terms. I must insist that the manuscript does not characterize any enzyme, so that it is senseless to talk about enzymatic activity throughout the paper.

CORRECTIONS MADE IN THE TEXT

In turn, the pattern of the catalysis is very abnormal, suggesting that the existence of an enzyme is unlikely (i.e. the showed the highest activity is obtained with a protein concentration of 3.62 μg/mL (see abstract), assuming that concentration is referred to total protein in the fungal extract). It is hard to explain why a greater amount of a supposed enzyme does not increase the activity. Catalytic system or reductant activity are the recommended terms for substituting the terms “enzyme” and “enzymatic activity”.

CORRECTIONS MADE IN THE TEXT

Reviewer 3 Report

Accept in present form

Author Response

Dear reviewer 3: Thank you for the comments and suggestions for the improvement of the research work.